# ABDUCTIVE REASONING OVER TEMPORAL KNOWLEDGE GRAPHS VIA LOGICAL HYPOTHESIS GENERATION

## ABSTRACT

Abductive reasoning (ABR) aims to infer plausible hypotheses that explain observed facts. Existing studies have mainly focused on abductive reasoning over static knowledge graphs, while the temporal setting remains underexplored. In this paper, we investigate **abductive reasoning on temporal knowledge graphs (ABTKG)** and propose a dedicated framework for this task. We first generate logical hypotheses that explain an observation at a given time, and then train a temporal hypothesis generator through supervised learning. However, supervision alone is insufficient to handle unfamiliar observations at specific time points. To address this limitation, we introduce a reinforcement learning objective defined on the TKG, which reduces the gap between the observation and the conclusion produced by the generated hypothesis. Experiments on four public TKG datasets demonstrate consistent improvements in both explanatory power and reasoning accuracy, with gains observed across all datasets.

## 1 INTRODUCTION

Reasoning is widely regarded as a fundamental component of human intelligence, encompassing induction, deduction, and abduction. Abductive reasoning seeks to infer plausible hypotheses that explain observed facts, and it has been applied across diverse domains. In law, it assists in reconstructing possible actions or motives. In epidemiology, it facilitates tracing the sources and transmission paths of infectious diseases across populations and regions. It also serves as a critical tool for scientific discovery and fault diagnosis. Moreover, temporal information provides additional context that can enhance both explanatory power and predictive accuracy.

Temporal knowledge graphs (TKGs) store time-stamped facts in the form of quadruples, thereby modeling how entities and relations evolve over time. Reasoning over TKGs leverages historical information to infer and predict new facts. Existing research has examined both interpolation, which focuses on recovering missing past events, and extrapolation, which aims at forecasting future events (Liu et al., 2023; Xu et al., 2023; Zhang et al., 2024). Other approaches utilize graph neural networks to encode temporal signals and capture long-range dependencies, thereby improving reasoning accuracy (Li et al., 2021; 2022; Chen et al., 2024).

Beyond predictive tasks, abductive reasoning offers an explanatory perspective for TKG reasoning. TKGs provide both structured representations and rich semantics, making them a natural foundation for ABR. The objective in this setting is to generate logical hypotheses that best explain an observed set of entities at a specific time while maintaining consistency with the evolving graph. Prior work has explored abductive reasoning on static knowledge graphs (Bai et al., 2024; Gao et al., 2025). In this paper, we extend this direction to the temporal domain. The integration of temporal cues not only enhances the explanatory quality of generated hypotheses but also ensures alignment with temporal evolution, thereby enriching structured knowledge and improving overall accuracy and reliability.

The abductive perspective is equally crucial in the context of temporal knowledge graphs (TKGs). Consider the first example in Figure 1, where observation $O_1$ consists of five well-known figures in technology companies. A corresponding logical hypothesis $H_1$ may state that, at a particular time, they are entrepreneurs, founders in the technology industry, and reside in the United States.

| Observations (O) | Hypotheses (H) | Interpretations |
|---|---|---|
| $O_1$={Bill Gates, Steve Jobs, Jeff Bezos, Michael Dell} | $H_1 = V_?$: $Occupation(V_?, Entrepreneur, 2002) \wedge LivedIn(V_?, USA, 2002)$ | These people are entrepreneurs, founders of technology companies, and were lived in the United States in 2002. |
| $O_2$={Harry Potter, The Lord of the Rings, Game of Thrones} | $H_2 = V_?$: $Genre(V_?, Fantasy, 2014) \wedge \neg Genre(V_?, Realistic, 2000) \wedge \neg Type(V_?, PictureBook, 2004)$ | These are works that belong to the fantasy genre as of 2014, have been adapted into films or television series, and are not classified as science fiction in 2000, realistic literature, or children's picture books as of 2004. |
| $O_3$={The Shawshank Redemption, Forrest Gump, Léon: The Professional, Pulp Fiction} | $H_3 = V_?$: $Movie(V_?, 1994) \wedge Genre(V_?, Drama, 1994) \wedge ReleaseDecade(V_?, 1990s)$ | These films all belong to the category of classic European and American movies of the 1990s. |

Figure 1: Examples of observations and the inferred logical hypotheses. Each case shows how a hypothesis explains the observation at a specific time.

The second example $O_2$ involves three books, where hypothesis $H_2$ explains their common genres. The third example $O_3$, drawn from the medical domain, concerns three diseases. The corresponding hypothesis $H_3$ indicates that these diseases are chronic, primarily affect adults, involve multiple complications, and are neither infectious nor cancers. From a general perspective, these cases illustrate that time-aware abductive reasoning can produce logical hypotheses that accurately explain temporal observations.

Despite its promise, this task remains highly challenging. Facts in TKGs evolve continuously, and temporal constraints in hypotheses are strongly shaped by the observed entities. Even minor variations in the observation can lead to substantially different hypotheses. For example, the statement "the United States visits the United Kingdom" refers to distinct events depending on the timestamp. In case $O_1$, if "Travis Kalanick" is included, the temporal assignments of the hypothesis may shift entirely. This demonstrates that observations are pivotal not only for hypothesis formulation but also for selecting the correct temporal scope.

To address these challenges, we propose **ABTKG**, a framework for abductive reasoning over temporal knowledge graphs. ABTKG is composed of three main components. First, we construct observation–hypothesis pairs from TKGs and train a temporal hypothesis generator, in which temporal information is fused with relations during data construction and training to constrain outputs. Second, under supervised learning, the generator produces logical hypotheses conditioned on the given observations. Third, to further improve accuracy and generalization, we incorporate reinforcement learning (RL) with feedback from the TKG. In particular, we adopt proximal policy optimization (PPO) to reduce the discrepancy between the observed evidence and the conclusion derived from the generated hypothesis. Unlike conventional RL-based abduction on static KGs, which often struggles under temporal sparsity, ABTKG introduces a tailored reward function specifically designed to accommodate temporal constraints.

We evaluate the proposed framework on three widely used benchmark datasets, ICEWS14, YAGO, and WIKI. The experimental results consistently demonstrate the effectiveness of ABTKG, highlighting its strong reasoning ability and robust generalization across different temporal knowledge graphs.

We extend the task of abductive reasoning from static knowledge graphs to the temporal setting, where the goal is to generate time-aware logical hypotheses that more accurately explain the given observations.

- To address temporal dynamics, we propose a time-aware generation framework that incorporates temporal signals into hypothesis construction and introduces an improved reward function to enhance both learning efficiency and output quality.

- We present the first systematic study of abductive reasoning over temporal knowledge graphs.

- Experiments on four widely used benchmark datasets demonstrate the effectiveness of our method, showing consistent improvements in explanatory power and reasoning accuracy across all benchmarks.

## 2 RELATED WORK

### 2.1 TEMPORAL KNOWLEDGE GRAPH REASONING

Temporal knowledge graph reasoning focuses on modeling how entities and relations evolve over time. Snapshot-based methods exploit historical subgraphs to predict future facts, as in RE-NET (Jin et al., 2020) with an encoder–aggregator, or CyGNet (Zhu et al., 2021) with a copy mechanism for recurring events. Other approaches, such as EvoKG (Park et al., 2022), RE-GCN (Li et al., 2021), and TiRGN (Li et al., 2022), capture subgraph evolution, temporal gates, and time embeddings to handle long-range dependencies. More recent work (e.g., HGLS (Zhang et al., 2023), SMiFY (Liu et al., 2023), CENET (Xu et al., 2023), PLEASING (Zhang et al., 2024), LSEN (Wang et al., 2024), HIP (He et al., 2024)) seeks to build global views, simplify architectures, or jointly learn short- and long-term patterns for improved reasoning accuracy.

Beyond snapshot-based models, several methods extend static KG techniques to temporal domains. Know-Evolve (Trivedi et al., 2017) and GHNN (Ju et al., 2022) employ temporal point processes, while TA-DistMult (Garcia-Duran et al., 2018) and TeMP (Wu et al., 2020) integrate temporal signals into embedding and message-passing frameworks. Explainable models such as xERTE (Han et al., 2020) further construct inference subgraphs to enhance interpretability. Overall, these methods differ in how they encode history (local vs. global), time (discrete vs. continuous), and structure (single-hop vs. multi-hop), and are complementary to the abductive perspective considered in this work.

### 2.2 ABDUCTIVE REASONING

Abductive reasoning aims to infer hypotheses that best explain an observation under background knowledge. In natural language inference, $\alpha$-NLI (Bhagavatula et al., 2020) introduces abductive commonsense reasoning, with follow-up studies improving performance or exploring non-standard settings (Qin et al., 2020; Kadiķis et al., 2022; Chan et al., 2023; Zhao et al., 2024). More recently, large language models have been tested on open-world abductive reasoning (Zhong et al., 2023; Del & Fishel, 2023; Thagard, 2024) and abstract reasoning tasks (Liu et al.; Zheng et al., 2025).

In neuro-symbolic learning, Abductive Learning (ABL) (Zhou, 2019) couples perception with logic: candidate facts are extracted by machine learning models and then corrected through symbolic abduction, with feedback improving the perception module. Extensions such as ARLC (Camposampiero et al., 2024) and ABL-Refl (Hu et al., 2025) enhance context awareness, error correction, and generalization, further strengthening the integration of learning and reasoning.

## 3 PROBLEM DEFINITION

A temporal knowledge graph (TKG) is defined as a sequence of snapshots $\mathcal{G} = \{G_1, G_2, \ldots, G_T\}$. Each snapshot $G_t = (V, R, E_t)$ represents the set of facts observed at time $t$, where $V$ is the entity set, $R$ is the relation set, and $E_t \subseteq V \times R \times V \times \{t\}$ is the set of quadruples. Each fact is written as $(u, r, v, t)$ with $u, v \in V$ and $r \in R$. We follow the open-world assumption (Drummond & Shearer, 2006), treating unobserved facts as *unknown* rather than *false*. The complete graph $\bar{\mathcal{G}}$ is hidden during training, and the observed graph $\mathcal{G}$ is a subset, i.e., $\mathcal{G} \subseteq \bar{\mathcal{G}}$. We denote $r(u, v, t) = $ true if $(u, r, v, t) \in \bar{\mathcal{G}}$.

Abductive reasoning on a TKG involves two components: an *observation $O$* and a *hypothesis $H$*. Given a snapshot $G_t$, the observation $O$ is defined as a set of entities $O = \{o_1, o_2, \ldots, o_n\}$ with $o_i \in V$. A hypothesis $H$ is a first-order logical query over $G_t$, composed of existential quantifiers and logical operators $\wedge$, $\vee$, and $\neg$. Without loss of generality, $H$ can be expressed in disjunctive normal form (DNF):

$$H(V_?) = \exists V_1, \ldots, V_k : e_{1,t} \vee \cdots \vee e_{\ell,t},$$
$$e_{i,t} = r_{i1,t} \wedge \cdots \wedge r_{im_i,t}, \tag{1}$$

where $\{V_1, \ldots, V_k\} \subseteq V$ are variables. Each literal $r_{ij,t}$ is either $r(u, v, t)$ or $\neg r(u, v, t)$, with $u, v$ chosen from the variables or any entity in $G_t$. Typical examples include $r(v, V, t)$, $\neg r(v, V, t)$, $r(V, V', t)$, and $\neg r(V, V', t)$. The semantics of $H$ on a graph $G$ is given by

$$[\![H]\!]_G = \{ V_? \in V \mid H|_G(V_?) = \text{true} \}. \tag{2}$$

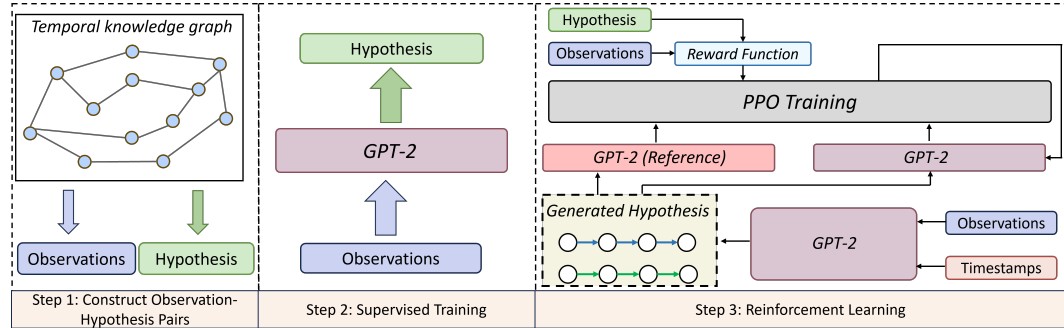

Figure 2: Overview of the training pipeline for temporal hypothesis generation. Step 1: sample diverse logical patterns and derive observations through graph search on the training TKG. Step 2: train a temporal hypothesis generator conditioned on both observations and time. Step 3: apply reinforcement learning with TKG feedback (PPO) to better align generated hypotheses with future observations.

The objective of abductive reasoning is to identify a hypothesis $H$ such that its conclusion on the hidden graph, $[\![H]\!]_{\bar{\mathcal{G}}}$, closely matches the observation $O$. We measure this similarity using the Jaccard index:

$$\text{Jaccard}([\![H]\!]_{\bar{\mathcal{G}}}, O) = \frac{|[\![H]\!]_{\bar{\mathcal{G}}} \cap O|}{|[\![H]\!]_{\bar{\mathcal{G}}} \cup O|}. \tag{3}$$

Hence, the goal is to find $H$ that maximizes $\text{Jaccard}([\![H]\!]_{\bar{\mathcal{G}}}, O)$.

## 4 METHOD

We present **ABRTKG**, a generative framework for abductive reasoning over temporal knowledge graphs. The overall pipeline is illustrated in Figure 2, and consists of three main components: (1) constructing temporal observation–hypothesis pairs, (2) training a temporal hypothesis generator with supervision, and (3) applying reinforcement learning with TKG feedback.

### 4.1 CONSTRUCTING TEMPORAL OBSERVATION–HYPOTHESIS PAIRS

In the first step, we build paired training data of observations and hypotheses using a set of predefined logical patterns. Specifically, we adopt thirteen patterns and allocate an equal number of samples to each. For a given pattern, we construct a hypothesis and query the training TKG to derive its conclusion, which serves as the observation. All steps are carried out under temporal constraints.

Formally, given a TKG $G$ and a logical pattern $P$, we select a node $v$ and recursively construct a hypothesis whose answer is $v$ and whose structure follows $P$. The recursion employs three operators: projection, intersection, and union. In practice, we merge relations and time by remapping each quadruple $(u, r, v, t)$ into a triple $(u, r_t, v)$. If the final operator is projection, we select an inbound edge $(u, r_t, v)$ and recurse on $u$ with a subpattern of $P$. If it is intersection, we recurse on both subpatterns anchored at the same $v$. If it is union, we recurse on one branch anchored at $v$ and on the other branch from any node, since only one branch must yield $v$ as the answer. The recursion terminates when the current node is an entity. Entity types are also attached during construction to enrich semantics.

### 4.2 TRAINING THE TEMPORAL HYPOTHESIS GENERATOR

In the second step, we train a conditional generative model using the constructed pairs. Let $\mathbf{o} = [o_1, \ldots, o_m]$ denote the observation tokens, $\mathbf{h} = [h_1, \ldots, h_n]$ the hypothesis tokens, and $\mathbf{t} = [t_1, \ldots, t_c]$ the time tokens. We optimize the conditional log-likelihood:

$$\log p_\theta(\mathbf{h} \mid \mathbf{o}, \mathbf{t}) = \sum_{i=1}^{n} \log p_\theta(h_i \mid \mathbf{o}, h_{1:i-1}, \mathbf{t}). \tag{4}$$

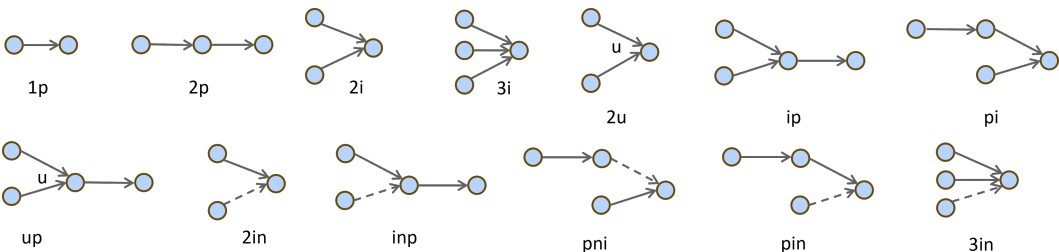

Figure 3: Thirteen types of logical hypothesis patterns.

The model $p_\theta$ is implemented as a Transformer decoder. Temporal information is injected by linearly mapping time and relation representations, merging them, and rewriting the corresponding tokens prior to decoding.

### 4.3 REINFORCEMENT LEARNING WITH TKG FEEDBACK

While supervised training enables the model to mimic target structures, it does not guarantee faithful explanations, especially for unseen observations at future timestamps. Small mismatches may also cause large reward fluctuations, leading to unstable optimization. To address these issues, we incorporate reinforcement learning (RL) with feedback from the TKG.

Let $G_{\text{train}}$ denote the observed training TKG, $\mathbf{o}$ the observation, and $\mathbf{h}$ the generated hypothesis. Let $O$ and $H$ represent their graph-level forms. We define three set-based rewards on $G_{\text{train}}$:

$$f_1 = \text{Jaccard}(\llbracket H \rrbracket_{G_{\text{train}}}, O), \quad f_2 = \text{Dice}(\llbracket H \rrbracket_{G_{\text{train}}}, O), \quad f_3 = \text{Overlap}(\llbracket H \rrbracket_{G_{\text{train}}}, O). \quad (5)$$

These metrics jointly provide smoother gradients and greater tolerance to minor mismatches, which is crucial when valid matches are sparse under temporal constraints.

Following (Ziegler et al., 2019), we treat the supervised model $p_\theta$ as the reference and initialize the policy $q$ with $p_\theta$. We optimize $q$ with proximal policy optimization (PPO), adding a KL penalty to keep the policy close to the reference:

$$\max_q \; \mathbb{E}_{\mathbf{o} \sim D, \, \mathbf{h} \sim q(\cdot|\mathbf{o},\mathbf{t})} \left[ \lambda_1 f_1 + \lambda_2 f_2 + \lambda_3 f_3 - \beta \log \frac{q(\mathbf{h}|\mathbf{o},\mathbf{t})}{p_\theta(\mathbf{h}|\mathbf{o},\mathbf{t})} \right], \quad (6)$$

where $D$ is the training distribution of observations, $\lambda_1, \lambda_2, \lambda_3$ are weighting coefficients, and $\beta$ is a dynamic penalty term (Ouyang et al., 2022). This objective encourages $q$ to generate coherent, time-aware hypotheses while remaining close to the supervised model.

## 5 EXPERIMENTS

### 5.1 DATASETS

We evaluate our method on four temporal knowledge graphs: ICEWS14 (Garcia-Duran et al., 2018), ICEWS18, YAGO (Mahdisoltani et al., 2014), and WIKI (Leblay & Chekol, 2018). Table 1 summarizes dataset statistics, including the numbers of entities, relations, timestamps, and edges. Following prior work, we split edges into training, validation, and test sets with an 8:1:1 ratio, and construct three graphs accordingly: $G_{\text{train}}$ (training edges only), $G_{\text{valid}}$ (training + validation), and $G_{\text{test}}$ (training + validation + test).

During sampling, each observation set is restricted to at most 32 entities for tractability. To increase difficulty, validation hypotheses are required to include at least one entity not appearing in $G_{\text{train}}$, and test hypotheses to include at least one entity not appearing in $G_{\text{valid}}$. This progressive design ensures more challenging evaluation settings. Following abductive reasoning studies on static KGs (Bai et al., 2024; Gao et al., 2025), we adopt thirteen predefined logical patterns. Eight patterns belong to the EPFO class without negation (1p, 2p, 2u, 3i, ip, up, 2i, pi), while the remaining five incorporate negation (2in, 3in, inp, pni, pin). Figure 3 illustrates their structures.

Table 1: Details of the four TKG datasets.

| Dataset | Relations | Entities | Timestamps | Training | Validation | Testing | All Edges |
|---------|-----------|----------|------------|----------|------------|---------|-----------|
| ICEWS14 | 230 | 7,128 | 365 | 139,936 | 17,492 | 17,492 | 174,920 |
| ICEWS18 | 256 | 23,033 | 304 | 749,692 | 93,712 | 93,712 | 937,116 |
| YAGO | 10 | 10,623 | 189 | 321,741 | 40,218 | 40,218 | 402,177 |
| WIKI | 24 | 12,435 | 232 | 1,071,894 | 133,988 | 133,986 | 1,339,868 |

## 5.2 EVALUATION METRICS

We assess semantic quality using set-based similarity between the hypothesis conclusion and the observation. Given a generated hypothesis $H$, we compute its conclusion $[\![H]\!]_{G_{\text{test}}}$ and evaluate Jaccard, Dice, and Overlap against the observation $O$. We further report Smatch (**?**), which measures structural similarity between the predicted hypothesis and the gold reference by comparing nodes, edges, and labels in their graph representations. Smatch is used as a reference metric, as exact structural matches are not required.

For computation, we augment each variable node with an `instance` edge to a dummy node. Given a predicted hypothesis $H_p$ and a gold hypothesis $H_g$, the Smatch score is defined as

$$S(H_p, H_g) = \text{Smatch}\big(G_A(H_p), G_A(H_g)\big), \tag{7}$$

where $G_A(\cdot)$ denotes the augmented graph representation.

## 5.3 EXPERIMENTAL SETTINGS

We use GPT-2 as the backbone for temporal hypothesis generation. For supervised training, we employ the AdamW optimizer with hyperparameters selected by grid search. The learning rate is set to $5 \times 10^{-5}$, the batch size is 256 for all datasets, and a 100-step linear warmup is applied. During reinforcement learning, we adopt proximal policy optimization with a dynamic KL penalty schedule (Ouyang et al., 2022). All experiments are conducted on NVIDIA A100 GPUs (80GB) using PyTorch.

Table 2: Detailed Jaccard scores over 13 query patterns; "Ave." is the average Jaccard. "Smatch Ave." is the average Smatch.

| Datasets | Model | 1p | 2p | 2i | 3i | ip | pi | 2u | up | 2in | 3in | pni | pin | inp | Ave. | Smatch Ave. |
|----------|-------|----|----|----|----|----|----|----|----|-----|-----|-----|-----|-----|------|-------------|
| ICEWS14 | GPT-2 | 0.684 | 0.669 | 0.845 | 0.863 | 0.683 | 0.834 | 0.660 | 0.654 | 0.756 | 0.770 | 0.795 | 0.733 | 0.727 | 0.744 | **0.541** |
| | +RL | **0.887** | **0.850** | **0.973** | **0.988** | **0.881** | **0.964** | **0.808** | **0.833** | **0.895** | **0.900** | **0.913** | **0.863** | **0.881** | **0.895** | 0.427 |
| ICEWS18 | GPT-2 | 0.578 | 0.506 | 0.566 | 0.650 | 0.572 | 0.527 | 0.623 | 0.469 | 0.563 | 0.583 | 0.574 | 0.483 | 0.408 | 0.546 | **0.549** |
| | +RL | **0.765** | **0.692** | **0.808** | **0.847** | **0.763** | **0.782** | **0.642** | **0.660** | **0.762** | **0.779** | **0.797** | **0.669** | **0.620** | **0.737** | 0.448 |
| YAGO | GPT-2 | 0.801 | 0.808 | 0.810 | 0.822 | 0.793 | 0.742 | **0.798** | 0.794 | 0.756 | 0.773 | 0.737 | 0.730 | 0.724 | 0.776 | **0.575** |
| | +RL | **0.847** | **0.828** | **0.906** | **0.910** | **0.844** | **0.837** | 0.783 | **0.803** | **0.860** | **0.861** | **0.834** | **0.761** | **0.772** | **0.834** | 0.478 |
| WIKI | GPT-2 | 0.761 | 0.732 | 0.803 | 0.813 | 0.766 | 0.787 | 0.811 | 0.709 | 0.684 | **0.693** | 0.682 | 0.630 | 0.634 | 0.731 | **0.540** |
| | +RL | **0.895** | **0.821** | **0.881** | **0.883** | **0.890** | **0.883** | **0.860** | **0.800** | **0.769** | 0.691 | **0.772** | **0.706** | **0.789** | **0.818** | 0.430 |

## 5.4 MAIN RESULTS

Table 2 presents results on the four datasets, reporting Jaccard scores across 13 query patterns. Overall, the GPT-2 generator effectively captures temporal signals from TKGs.

On ICEWS14, reinforcement learning (RL) produces clear improvements on complex patterns such as multi-hop and union, though slight declines are observed on a few simple cases (e.g., 3i and pi). On ICEWS18, we observe modest average gains, with RL enhancing performance on 2i, 3i, 2in, 3in, and pni, while simple patterns (1p, 2p, ip) show minor decreases. On YAGO, RL generally reduces scores across multiple patterns, particularly 2i, 3i, ip, and pi. This indicates that in structurally rich graphs where the supervised model already generalizes well, RL may introduce additional variance

rather than consistent improvements. On WIKI, the largest dataset, performance on simple patterns remains stable, while RL yields noticeable gains on complex cases (e.g., 2in). In summary, RL proves most effective for complex logical queries—particularly those involving negation, intersection, and union—whereas its benefit on simple patterns is limited.

Table 3: Effect of explicitly concatenating timestamps on ICEWS14 and ICEWS18. Scores are Jaccard over 13 query patterns; "Ave." is the average.

| Datasets | Model | 1p | 2p | 2i | 3i | ip | pi | 2u | up | 2in | 3in | pni | pin | inp | Ave. |
|---|---|---|---|---|---|---|---|---|---|---|---|---|---|---|---|
| ICEWS14 | Mix | 0.684 | **0.669** | **0.845** | **0.863** | **0.683** | **0.834** | **0.660** | **0.654** | **0.756** | **0.770** | **0.795** | **0.733** | **0.727** | **0.744** |
| | Explicit Timestamps | **0.770** | 0.623 | 0.491 | 0.387 | 0.461 | 0.337 | 0.506 | 0.397 | 0.592 | 0.574 | 0.511 | 0.473 | 0.514 | 0.510 |
| | Mix+RL | **0.887** | **0.850** | **0.973** | **0.988** | **0.881** | **0.964** | **0.808** | **0.833** | **0.895** | **0.900** | **0.913** | **0.863** | **0.881** | **0.895** |
| | Explicit Timestamps | 0.767 | 0.638 | 0.502 | 0.368 | 0.510 | 0.332 | 0.513 | 0.427 | 0.587 | 0.550 | 0.497 | 0.461 | 0.535 | 0.514 |
| ICEWS18 | Mix | 0.578 | 0.506 | **0.566** | **0.650** | **0.572** | **0.527** | **0.623** | **0.469** | 0.563 | 0.583 | **0.574** | **0.483** | **0.408** | **0.546** |
| | Explicit Timestamps | **0.802** | **0.551** | 0.479 | 0.444 | 0.443 | 0.286 | 0.622 | 0.352 | **0.656** | **0.642** | 0.512 | 0.410 | 0.368 | 0.505 |
| | Mix+RL | **0.765** | **0.692** | **0.808** | **0.847** | **0.763** | **0.782** | **0.642** | **0.660** | **0.762** | **0.779** | **0.797** | **0.669** | **0.620** | **0.737** |
| | Explicit Timestamps | 0.796 | 0.531 | 0.507 | 0.469 | 0.421 | 0.289 | 0.621 | 0.340 | 0.666 | 0.653 | 0.536 | 0.422 | 0.334 | 0.507 |

## 5.5 EXPLICIT TEMPORAL INFORMATION

We investigate the impact of incorporating explicit temporal signals on ICEWS14 and ICEWS18 (Table 3).Concatenating timestamp tokens substantially reduces performance: the average Jaccard score decreases from 0.680 to 0.510 on ICEWS14, and from 0.546 to 0.505 on ICEWS18. The degradation is more pronounced on complex patterns such as intersections (2i, 3i) and mixed types (ip, pi). This outcome can be attributed to the sparse and discrete nature of timestamps, where naive concatenation introduces noise and weakens the model's ability to capture entity–relation dynamics. Without structured temporal modeling, the generator tends to rely on surface matching and fails to exploit long-range dependencies. These findings suggest that temporal information should be integrated in a structured or implicit manner rather than through direct concatenation.

We further assess explicit timestamp tokens under reinforcement learning. The trend remains consistent: explicit temporal signals do not yield improvements, even when RL is applied. On ICEWS14, the average Jaccard score with RL drops from 0.895 (without explicit tokens) to 0.514 (with explicit tokens), and on ICEWS18 it drops from 0.737 to 0.507. Although minor gains are observed on a few simple patterns, they are small and inconsistent. We attribute this to a conflict between discrete timestamp tokens and the RL signal: while the tokens expand the input space and divert attention from structural cues, the reward primarily evaluates set-level answer quality rather than exact time tokens. In practice, implicit or structured temporal encodings—such as learned temporal embeddings, relative-time offsets, or gated conditioning—prove more reliable than appending raw timestamps.

Table 4: Ablation on reward functions for reinforcement learning. Scores are Jaccard over 13 query patterns; "Ave." is the average. $f_1$: Jaccard, $f_2$: Dice, $f_3$: Overlap.

| Datasets | Reward | 1p | 2p | 2i | 3i | ip | pi | 2u | up | 2in | 3in | pni | pin | inp | Ave. |
|---|---|---|---|---|---|---|---|---|---|---|---|---|---|---|---|
| ICEWS14 | $f_1$ | 0.858 | 0.834 | 0.952 | 0.962 | 0.857 | 0.949 | 0.728 | 0.817 | 0.883 | 0.885 | 0.913 | 0.854 | 0.869 | 0.874 |
| | $f_1 + f_2$ | 0.879 | 0.848 | 0.926 | 0.877 | 0.875 | 0.927 | 0.741 | 0.835 | 0.897 | 0.897 | **0.925** | 0.868 | 0.879 | 0.875 |
| | $f_1 + f_3$ | 0.872 | 0.848 | 0.929 | 0.906 | 0.869 | 0.932 | 0.766 | **0.836** | **0.902** | **0.904** | 0.924 | **0.873** | **0.882** | 0.880 |
| | $f_1 + f_2 + f_3$ | **0.887** | **0.850** | **0.973** | **0.988** | **0.881** | **0.964** | **0.808** | 0.833 | 0.895 | 0.900 | 0.913 | 0.863 | 0.881 | **0.895** |
| YAGO | $f_1$ | 0.883 | 0.822 | 0.860 | 0.760 | 0.728 | 0.695 | 0.772 | 0.731 | 0.781 | 0.712 | 0.671 | 0.686 | 0.693 | 0.753 |
| | $f_1 + f_2$ | 0.858 | 0.815 | 0.878 | 0.884 | 0.867 | 0.812 | 0.776 | 0.790 | 0.827 | 0.834 | 0.808 | 0.734 | **0.790** | 0.821 |
| | $f_1 + f_3$ | **0.893** | 0.821 | 0.864 | 0.865 | **0.880** | 0.817 | **0.794** | **0.805** | 0.840 | 0.838 | 0.825 | 0.736 | 0.781 | 0.827 |
| | $f_1 + f_2 + f_3$ | 0.847 | **0.828** | **0.906** | **0.910** | 0.844 | **0.837** | 0.783 | 0.803 | **0.860** | **0.861** | **0.834** | **0.761** | 0.772 | **0.834** |

## 5.6 EFFECT OF REWARD COMBINATIONS

We examine the impact of different reward configurations on ICEWS14 and YAGO (Table 4). The results demonstrate that combining multiple reward signals yields the best overall performance, highlighting the complementary nature of Jaccard, Dice, and Overlap.

On ICEWS14, using $f_1$ (Jaccard) alone is relatively stable but weaker on long-chain queries such as 2u and up. Adding $f_2$ (Dice) improves performance on several patterns, including 2p and 2in, though the average gain is limited and performance on 3i declines. Introducing $f_3$ (Overlap) provides further improvements, particularly on complex queries such as 3i, pi, and inp.

On YAGO, $f_1$ alone performs poorly on union and negation queries (e.g., 3in, pni, pin). Adding $f_2$ brings consistent improvements, while the full combination $f_1 + f_2 + f_3$ achieves the best results across most metrics, yielding approximately 8%–10% higher average scores compared with single-reward setups.

In summary, multi-signal reward design improves consistency and robustness across diverse query types, and provides more reliable guidance for learning complex reasoning patterns.

Table 5: Comparison of pretrained language models on temporal abductive reasoning. Scores are Jaccard over 13 query patterns; "Ave." is the average across patterns.

| Datasets | Model | 1p | 2p | 2i | 3i | ip | pi | 2u | up | 2in | 3in | pni | pin | inp | Ave. |
|---|---|---|---|---|---|---|---|---|---|---|---|---|---|---|---|
| ICEWS14 | GPT-2 | 0.608 | 0.599 | 0.769 | 0.804 | 0.606 | 0.761 | 0.603 | 0.593 | 0.698 | 0.743 | 0.730 | 0.670 | 0.654 | 0.680 |
| | +RL | **0.887** | **0.850** | **0.973** | **0.988** | **0.881** | **0.964** | **0.808** | **0.833** | **0.895** | **0.900** | **0.913** | **0.863** | **0.881** | **0.895** |
| | T5 | 0.411 | 0.397 | 0.449 | 0.477 | 0.410 | 0.465 | 0.609 | 0.391 | 0.497 | 0.549 | 0.520 | 0.468 | 0.404 | 0.465 |
| | +RL | 0.661 | 0.613 | 0.719 | 0.758 | 0.664 | 0.716 | 0.648 | 0.586 | 0.592 | 0.634 | 0.608 | 0.568 | 0.562 | 0.641 |
| YAGO | GPT-2 | 0.801 | 0.808 | 0.810 | 0.822 | 0.793 | 0.742 | **0.798** | **0.794** | 0.756 | 0.773 | 0.737 | 0.730 | 0.724 | 0.776 |
| | +RL | **0.858** | **0.815** | **0.878** | **0.884** | **0.867** | **0.812** | 0.776 | 0.790 | **0.827** | **0.834** | **0.808** | **0.734** | **0.790** | **0.821** |
| | T5 | 0.351 | 0.314 | 0.259 | 0.262 | 0.363 | 0.275 | 0.188 | 0.300 | 0.284 | 0.269 | 0.243 | 0.256 | 0.297 | 0.282 |
| | +RL | 0.613 | 0.461 | 0.457 | 0.470 | 0.601 | 0.477 | 0.341 | 0.445 | 0.447 | 0.429 | 0.382 | 0.358 | 0.456 | 0.457 |

## 5.7 EFFECT OF PRETRAINED LANGUAGE MODELS

We analyze the influence of different pretrained language models on temporal abductive reasoning, comparing a decoder-only architecture (GPT-2) with an encoder–decoder architecture (T5) on ICEWS14 and YAGO, both with and without reinforcement learning (Table 5).

On ICEWS14, GPT-2 consistently outperforms T5. The T5 baseline is particularly weak on multi-hop queries such as 2i, 3i, and inp. Although reinforcement learning improves T5, GPT-2 remains superior, leveraging historical patterns and reward signals to achieve higher scores across most query types. On YAGO, a similar trend is observed: GPT-2 exhibits stable performance with or without RL, while T5 yields lower overall scores and further degrades on certain patterns, including 2u, 3in, and pni.

These results indicate that, under our setting, encoder–decoder architectures struggle to capture complex temporal and logical dependencies. In contrast, GPT-2 proves to be a more effective backbone for temporal abductive reasoning, particularly when augmented with reinforcement learning. This advantage likely stems from the task's formulation as conditional sequence generation with long contexts, a setting where decoder-only models excel. T5 may require stronger temporal conditioning or constrained decoding strategies to narrow this gap.

## 6 CONCLUSION

In this work, we investigated abductive reasoning over temporal knowledge graphs. We proposed a time-aligned hypothesis generation framework that produces time-stamped logical hypotheses, thereby alleviating graph incompleteness. To further enhance performance, we incorporated rein-

forcement learning with feedback from the TKG, enabling a generator that balances explanatory quality with inference efficiency. Extensive experiments on multiple datasets demonstrate consistent improvements across evaluation metrics, validating the effectiveness of coupling temporal constraints with reinforcement learning for temporal abductive reasoning. Future work will explore integrating LLMs with structured temporal reasoning modules and testing on larger-scale open-world datasets.

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

## A  MORE EXPERIMENTAL DETAILS

In all experiments, the learning rate for supervised training is set to $1 \times 10^{-5}$ with a batch size of 256. We train for 30 epochs in the supervised stage. During reinforcement learning, we use a smaller batch size of 32 and sample 4 candidate answers per group. The hyperparameters are $\lambda_1 = 1.0$, $\lambda_2 = 0.2$, $\lambda_3 = 0.1$, and $\beta = 0.1$.

To avoid semantic leakage from pre-trained language models, we represent entities and relations by their IDs in the knowledge graph. When generating segments that contain new relations or timestamps, we synchronously produce the corresponding ID sequences. Under this setting, the task is essentially sequence or graph generation rather than a traditional natural language processing task. We therefore adopt an autoregressive GPT-2 architecture and find it well suited to this formulation.

## B  OBSERVATION–HYPOTHESIS PAIRS ON TEMPORAL KGS

In a temporal knowledge graph, an observation set $O$ may consist of several entity names, e.g., $O = \{$Harry Potter, The Lord of the Rings, Game of Thrones$\}$. Given $O$, an abductive reasoner should produce a logical hypothesis that best explains the observation, e.g.,

$$H_2(V_?) = \text{Genre}(V_?, \text{Fantasy}, 2014) \land \neg \text{Genre}(V_?, \text{Realistic}, 2000) \land \neg \text{Type}(V_?, \text{PictureBook}, 2004).$$

In natural language: these works belong to the fantasy genre in 2014; they do not belong to the realistic genre in 2000; and they are not classified as children's picture books in 2004.

## C  DETAILS OF USING SMATCH TO EVALUATE STRUCTURAL DIFFERENCES

The Smatch metric was designed for AMR graphs and requires that each variable node has exactly one instance edge that binds it to a concept. Our hypothesis graphs do not enforce this constraint, so direct computation would be unfair. We therefore perform a minimal conversion: for every variable node, we add a unified instance edge to a virtual node $v'$, which makes the graph compatible with AMR conventions and allows Smatch to stably measure structural differences.

Concretely, given a hypothesis $H$ and its graph $G(H)$, we build an AMR-friendly graph $G_A(H)$ as follows: (i) initialize $G_A(H) = G(H)$; (ii) add a virtual node $v'$ and the relation type instance; (iii) for each variable node $v \in G(H)$, add the edge instance$(v, v')$. We then define the structural similarity between a predicted hypothesis $H_p$ and a gold hypothesis $H_g$ as

$$S(H_p, H_g) \;=\; \mathrm{Smatch}\big(G_A(H_p),\; G_A(H_g)\big).$$

Under this conversion, alignment of variable nodes is anchored by their instance edges, after which other relation or attribute edges are compared. This procedure does not alter the search for the optimal variable-node mapping and yields a stable, leak-free similarity score between hypothesis graphs.

---

**Observations 1:** Barack Obama,  Vincent Gray,  Oleg Ostapenko
**Hypothesis 1:** $V_?$: *Host_a_visit($V_?$, Shamshad_Akhtar, 248)∧ Praise_or_endorse($V_?$ , Akhtar, 248)*
**Interpretations 1:** The set of subjects who, at time 248, and hosted a visit to **Shamshad Akhtar or** praised/endorsed **Akhtar** (counted once even if both hold).
**Conclusion 1:** Barack Obama,  Vincent Gray,  Oleg Ostapenko
**Jaccard Score 1:** 1.0

- - - - - - - - - - - - - - - - - - - - - - - - - - - - - - - - - - - - - - - - -

**Observations 2:** Mohammad Javad Zarif, Dmitry Mezentsev, Foreign Affairs (Brunei)
**Hypothesis 2:** $V_?$: *Make_optimistic_comment ($V_?$, Mohammad_Javad_Zarif, 58) ∧ Make_statement ($V_?$ , Mohammad_Javad_Zarif, 96)*
**Interpretations 2:** The set of subjects who, at time 58, **made an optimistic comment** about **Mohammad Javad Zarif and**, at time 96, **made a statement** about him.
**Conclusion 2:** Mohammad Javad Zarif, Dmitry Mezentsev, Foreign Affairs (Brunei)
**Jaccard Score 2:** 1.0

Figure 4: Case study of ABRTKG.

## D  CASE STUDY

To examine the model's reasoning behavior, we present two representative outputs in Fig. 4. In temporal KG abduction, the goal is to infer a time-stamped hypothesis that explains the given observations. Both cases show that our model can generate plausible temporal hypotheses aligned with the observations and their timestamps. The first case illustrates how the model uses logical operations (e.g., union and conjunction) to cover alternative explanations; the second case shows how it links events across timestamps to produce a consistent conclusion. Overall, these examples indicate stable controllability of the logical form and temporal consistency.

## E  EXPLICIT TIMESTAMP CONCATENATION

During sampling, we enumerate relation IDs and timestamp IDs and remap them into time-aware relation IDs $r_t$ to construct hypotheses. By "explicit timestamps," we mean directly appending timestamp tokens to the observation sequence to guide the generator toward time-constrained hypotheses. However, on ICEWS14 and ICEWS18 this strategy consistently reduces performance. Timestamps are sparse and discrete; naive concatenation injects noise, distracts the model from learning entity–relation dynamics, and weakens its ability to capture long-range temporal dependencies. These

results suggest that temporal signals should be modeled in a more structured or implicit manner (e.g., via time-aware relations or learned temporal encoders) rather than by simple concatenation.

