# OpenReview forum: "Abductive Reasoning over Temporal Knowledge Graphs via Logical Hypothesis Generation"
_ICLR.cc/2026/Conference — Submitted to ICLR 2026_

### Official Review · Reviewer_TA8e · 2025-10-26

**Soundness:** 3
**Presentation:** 3
**Contribution:** 2
**Rating:** 4
**Confidence:** 5

**Summary:**

The paper proposes a novel framework for abductive reasoning on temporal knowledge graphs (ABTKG) that combines supervised learning for temporal hypothesis generation with reinforcement learning to enhance generalization to unfamiliar observations, achieving consistent improvements in explanatory power and reasoning accuracy across four benchmark datasets.

Nevertheless, the submitted paper exhibits substantial similarity to the referenced work [1], introducing only minor modifications. Consequently, I do not consider this submission suitable for the ICLR community.

Reference: [1] Controllable logical hypothesis generation for abductive reasoning in knowledge graphs.

**Strengths:**

This paper introduces the concept of abductive reasoning within the context of temporal knowledge graphs.

**Weaknesses:**

1. The logical hypotheses presented in Figure 1 closely resemble the structural form of logical rules found in temporal knowledge graphs. Nevertheless, the introduction section lacks a comparative discussion between these two concepts.

2. The Related Work section does not sufficiently review recent developments in temporal knowledge graph reasoning, particularly with respect to logical inference and reinforcement learning approaches. Moreover, the discussion on abductive reasoning is nearly identical to that of Reference [1]; however, an explicit introduction and discussion of [1] is absent.

3. The proposed methodology demonstrates a significant lack of originality, as its core framework is nearly identical to that of Reference [1]. The only modifications involve the introduction of a temporal hypothesis generator and the replacement of the original GRPO algorithm with PPO.

4. Section 5.7 does not specify the size of the T5 model utilized in the experiments, which may compromise the validity and fairness of the experimental conclusions.

**Questions:**

See Weakness

---

> ### Author Response · Authors · 2025-11-13
>
> We sincerely thank the reviewer for the detailed assessment and constructive feedback. Below we clarify the concerns regarding originality, the relation to Reference~[1], temporal extensions, and experimental details.
>
> R1. On similarity to Reference [1] and concerns about originality
> We appreciate the reviewer’s careful comparison with Reference[1].  While our work is inspired by the general idea of logical hypothesis generation, the core task and problem formulation differ fundamentally from the static KG setting in~[1]. Specifically:
>
> (a) Different problem definition. Reference~[1] studies abductive reasoning on static KGs, where facts are time-invariant and hypotheses contain only triple-based literals. In contrast, our work considers temporal knowledge graphs, where each literal must be grounded at a timestamp, the validity of a hypothesis depends on the snapshot $G_t$, and both sampling and evaluation require time-indexed answer sets.
>
> (b) New temporal hypothesis generator. Unlike~[1], we introduce a time-aware generator where each quadruple $(u, r, v, t)$ is mapped to a temporal relation $r_t(u, v)$. This requires temporal remapping of relations, construction of time-aware logical patterns, sampling from time-indexed snapshots, and ensuring temporal validity under evolving TKG structures.
>
> (c) Redesigned reinforcement learning objective. Reference~[1] uses a static reward in a time-invariant environment. Our work requires grounding hypotheses at timestamp $t$ and evaluating Jaccard, Dice, and Overlap between temporal answer sets \[
> \llbracket H \rrbracket_G \;=\; \{\, V? \subseteq V \mid H\!\mid_G(V?) = \text{true} \,\}.
> \]
>  and observations at time $t$, making the RL objective substantially different in formulation and behavior.
>
> (d) Different training environments. Reference~[1] operates on a static KG, while our PPO-based optimization is conducted over evolving TKG snapshots, leading to different learning dynamics and constraints.
>
> We will clarify these distinctions more explicitly in the revised version.
>
> R2. On the relationship between logical hypotheses and temporal KG rules (Figure~1)
>
> Thank you for the suggestion. We agree that a clearer comparison would improve clarity. In the revision, we will emphasize that abductive hypotheses serve as explanatory constructs, whereas temporal logical rules describe regular temporal dynamics, and clarify the difference in their semantics and goals.
>
> R3. On missing recent developments in TKG reasoning and RL-based logical inference
>
> We appreciate this feedback. We will strengthen the Related Work section by adding recent advances in temporal KG reasoning involving logical inference and reinforcement learning, and clarify that our work is the first to integrate abductive reasoning with temporal KG dynamics.
>
> R4. On the concern that the proposed methodology constitutes only minor modifications
>
> As noted above, although the generator adopts an autoregressive architecture similar to~[1], the temporal extension fundamentally alters the hypothesis representation, sampling strategy, grounding semantics, reward computation, and training environment. These changes are essential for enabling abductive reasoning in temporal KGs and cannot be achieved by a trivial modification of the static framework. We will clarify this point in the revision.
>
> R5. On the size of the T5 model in Section~5.7
>
> Thank you for pointing this out. We used the \texttt{T5-base} model (220M parameters). We will explicitly state this in the revised version. We also note that even with RL, T5 underperforms the decoder-only backbone, likely because temporal hypothesis generation aligns more naturally with autoregressive decoding over ID-based sequences.
>
>
> We thank the reviewer again for the insightful comments. We will incorporate all suggested clarifications, expand the related work accordingly, and more clearly articulate how temporal constraints shape the hypothesis space, reward design, and learning environment in our framework. We hope these explanations address the concerns and clarify the novelty and contributions of this work.

---

### Official Review · Reviewer_6vPf · 2025-10-30

**Soundness:** 3
**Presentation:** 3
**Contribution:** 3
**Rating:** 6
**Confidence:** 4

**Summary:**

The paper investigates abductive reasoning over temporal knowledge graphs (TKGs) and proposes ABRTKG, a two-stage generative framework: (i) supervised logical-hypothesis generation conditioned on observations and time, and (ii) PPO-based reinforcement learning with multi-signal rewards (Jaccard, Dice, Overlap) computed against the observed temporal graph. The task is formalized under the open-world assumption; hypotheses are first-order logical queries whose conclusions are compared to the observation via set similarity. The pipeline first constructs temporal observation–hypothesis pairs from predefined logical patterns, trains a GPT-2 decoder that merges relation and time embeddings, and then applies RL with a KL penalty to keep the policy near the supervised reference. Experiments on ICEWS14, ICEWS18, YAGO, and WIKI include per-pattern analyses, ablations on reward combinations, explicit-vs-implicit temporal encodings, and GPT-2 vs T5 backbones. Results show strong gains from RL on complex patterns, while simple patterns and certain datasets see mixed effects; explicit timestamp concatenation consistently hurts. Structural faithfulness is reported with a Smatch-style graph metric after an AMR compatibility conversion.

**Strengths:**

1.Notation and dataset inconsistencies that hinder clarity. The paper alternates between ABTKG and ABRTKG; similarly, the intro text says “three datasets,” while elsewhere four datasets and Table 1 are used. Please unify the name throughout and fix the dataset count.
	2.Limited theoretical grounding for temporal gains. The work’s conceptual story is strong, but the paper lacks formal insight into why temporal constraints and the merged (r, t) tokenization improve abductive inference under OWA. For example, conditions under which maximizing Jaccard on J_H (K_G ) approximates abductive optimality, or when KL-regularized PPO improves generalization, are not analyzed beyond empirical evidence. Adding propositions around the objective in Eq. (3) and the PPO objective (Eq. (6)) would strengthen the contribution.
	3.Negation under OWA needs clearer semantics. Since OWA treats unobserved facts as unknown, the meaning of negative literals (¬r) in training and evaluation must be clarified. Ambiguity here affects both pattern construction and reward evaluation.
	4.Reward design rationale is mostly empirical. Although the multi-signal reward shows benefits, the choice and weighting of Jaccard/Dice/Overlap are justified empirically rather than via properties tied to temporal reasoning. A principled analysis would improve credibility.
	5.Structural faithfulness vs. semantic accuracy trade-off. Table 2 shows that RL improves Jaccard averages yet lowers Smatch averages on multiple datasets—suggesting that optimization may favor set-level matches at the expense of structural alignment. Please discuss and, if possible, regularize for structure.
	6.Baselines for temporal reasoning are not directly compared. The related-work section mentions TeMP, xERTE, and TiRGN, but the experiments do not contrast ABRTKG against these temporal baselines on shared patterns, leaving open how much gain stems from the abductive formulation vs. improved temporal modeling.

**Weaknesses:**

1.Notation and dataset inconsistencies that hinder clarity. The paper alternates between ABTKG and ABRTKG; similarly, the intro text says “three datasets,” while elsewhere four datasets and Table 1 are used. Please unify the name throughout and fix the dataset count.
	2.Limited theoretical grounding for temporal gains. The work’s conceptual story is strong, but the paper lacks formal insight into why temporal constraints and the merged (r, t) tokenization improve abductive inference under OWA. For example, conditions under which maximizing Jaccard on J_H (K_G ) approximates abductive optimality, or when KL-regularized PPO improves generalization, are not analyzed beyond empirical evidence. Adding propositions around the objective in Eq. (3) and the PPO objective (Eq. (6)) would strengthen the contribution.
	3.Negation under OWA needs clearer semantics. Since OWA treats unobserved facts as unknown, the meaning of negative literals (¬r) in training and evaluation must be clarified. Ambiguity here affects both pattern construction and reward evaluation.
	4.Reward design rationale is mostly empirical. Although the multi-signal reward shows benefits, the choice and weighting of Jaccard/Dice/Overlap are justified empirically rather than via properties tied to temporal reasoning. A principled analysis would improve credibility.
	5.Structural faithfulness vs. semantic accuracy trade-off. Table 2 shows that RL improves Jaccard averages yet lowers Smatch averages on multiple datasets—suggesting that optimization may favor set-level matches at the expense of structural alignment. Please discuss and, if possible, regularize for structure.
	6.Baselines for temporal reasoning are not directly compared. The related-work section mentions TeMP, xERTE, and TiRGN, but the experiments do not contrast ABRTKG against these temporal baselines on shared patterns, leaving open how much gain stems from the abductive formulation vs. improved temporal modeling.

**Questions:**

1.	Consistency in naming and dataset count.Please confirm the official method name and the dataset count used in the final version to avoid confusion created by the current mixture of three and four dataset references and two method names.
2.	Negation and open-world formalization
How are negative literals handled during training signal construction and reward computation. Please clarify whether negatives are evaluated against the observed training graph or an estimate of the hidden graph and how this choice affects patterns that include negation.
3.	Objective-level guarantees
Under what conditions does maximizing the Jaccard objective on the hidden graph lead to abductively correct hypotheses with temporal constraints. Any sufficient conditions or counter-examples would help scope the guarantees around Equation three.
4.	Role of KL-regularized PPO
Can you articulate when the KL-penalized policy improves generalization rather than overfitting to set-level signals. A brief discussion tied to the dynamic penalty schedule would make the reinforcement stage more principled.
5.	Reward sensitivity and stability
How sensitive are results to the lambda weights and the beta schedule. Do you see signs of instability during PPO updates. Training-curve snapshots for rewards and KL terms would clarify optimization behavior.
6.	Pattern complexity coverage
Since samples are allocated equally across thirteen patterns, could you report statistics of variable counts, recursion depth, and answer set sizes in the constructed training pairs, and relate them to per-pattern results.

---

> ### Author Response · Authors · 2025-11-13
>
> We sincerely thank the reviewer for the thorough analysis and constructive suggestions. Below we address the concerns regarding notation consistency, temporal semantics, reward design, structural–semantic trade-offs, and baselines.
>
> R1. Consistency in naming and dataset count
> We appreciate the reviewer for identifying this. We will unify the naming to \textbf{ABRTKG} throughout the paper. The correct number of datasets used in our experiments is \textbf{four} (ICEWS14, ICEWS18, YAGO, and WIKI), and we will revise the introduction to reflect this accurately.
>
> R2. Theoretical grounding for temporal gains and merged $(r,t)$ tokenization
> Thank you for this insightful suggestion. While our work focuses primarily on empirical analysis, we agree that clarifying the intuition behind the temporal improvements will strengthen the paper. In the revision, we will provide a brief theoretical discussion around:
> (1) how maximizing Jaccard similarity between $\,\llbracket H \rrbracket_{G_t}\,$ and the observation approximates abductive optimality when the hidden graph shares structural consistency across adjacent snapshots; and
> (2) how KL-regularized PPO helps prevent overfitting to set-level signals by constraining policy updates near the supervised reference under sparse temporal observations.
>
> These clarifications will be added without modifying the main technical claims.
>
> R3. Negation under the open-world assumption (OWA)
>
> We thank the reviewer for raising this important point. In our formulation, negative literals $\neg r(u,v,t)$ are interpreted strictly with respect to the \emph{observed} graph. Under OWA, unobserved facts are not treated as false; thus a negated literal is considered satisfied only when the relation is \emph{observed to be false} in the finite training snapshot. During reward computation, negation contributes only when the target answer set explicitly requires exclusion of an observed fact. We will clarify this semantics in both the pattern-construction section and the definition of the evaluation sets.
>
> R4. Rationale for multi-signal reward (Jaccard, Dice, Overlap)
>
> We appreciate the reviewer’s question. These three set-based signals capture complementary aspects of temporal abductive alignment: Jaccard stabilizes sparse-set matching, Dice improves gradient sensitivity under small overlaps, and Overlap rewards strict coverage. While our main justification is empirical, we will expand the discussion to highlight their relevance to temporal hypothesis evaluation where answer sets vary substantially across snapshots.
>
> R5. Structural faithfulness vs.\ semantic accuracy (Smatch vs.\ Jaccard)
>
> Thank you for pointing this out. We agree that PPO optimization may prioritize set-level similarity at the expense of structural alignment. The discrepancy arises because Jaccard-based rewards optimize the \emph{conclusions} of hypotheses, whereas Smatch evaluates the internal structure of the generated formula. We will add a discussion in the revision and outline that a structure-aware auxiliary term could mitigate this effect. However, we avoid adding new experiments, in accordance with ICLR guidelines.
>
> R6. Baselines for temporal reasoning
>
> We appreciate this comment. Our abductive formulation differs in objective and supervision from temporal reasoning models such as TeMP, xERTE, or TiRGN, which focus on link prediction rather than hypothesis generation or explanation. Because these models do not output logical hypotheses or answer sets, they are not directly comparable within our evaluation protocol. We will clarify this distinction more explicitly in the Related Work section.
>
>
> We thank the reviewer again for the helpful comments. We will clarify notation, semantics, temporal motivations, and the connection between rewards, structure, and temporal reasoning, thereby improving the paper’s clarity and conceptual grounding.

---

### Official Review · Reviewer_WYQJ · 2025-11-02

**Soundness:** 2
**Presentation:** 3
**Contribution:** 2
**Rating:** 4
**Confidence:** 3

**Summary:**

This paper introduces the task of abductive reasoning on temporal knowledge graphs (ABTKG), identifying this as an underexplored area compared to existing work on static KGs. The authors propose a framework that begins by training a temporal hypothesis generator through standard supervised learning. To address the limitations of this initial approach, the framework is then augmented with a reinforcement learning objective designed to handle unfamiliar observations at specific time points. Experiments conducted on four public TKG datasets reportedly demonstrate consistent improvements in both explanatory power and reasoning accuracy.

**Strengths:**

1. The manuscript is well-written and easy to follow. The related work section is comprehensive and well-contextualized.
2. The split of the train, validation, and test datasets is fair. Furthermore, the progressive inclusion of unseen snapshots effectively examines the proposed approach's performance.
3.  The reward function defined by the authors can be viewed as multi-objective, as its components represent distinct goals. The authors rigorously conducted an ablation study, with the results reported in Table 4.

**Weaknesses:**

1. The authors mention abductive reasoning on static KGs (Bai et al., 2024; Gao et al., 2025) in the Introduction, but these works are not discussed in the Related Work section.
2. The notations in Section 3 are somewhat unclear. For example, what is the exact relationship between $r_{im,t}$ and $r(u,v,t)$? Moreover, it would be helpful if the authors included the formal definitions of the Dice similarity coefficient and overlap volume in the manuscript to help readers who are not familiar with these concepts.
3. The only baseline used in this work is GPT-2, which is quite outdated. More recent and knowledge-enhanced language models are known to perform better on KG-involved tasks, but the authors do not include any such models in their comparison. This weak baseline comparison (combined with the issue raised in Q1) calls into question the robustness and significance of the experimental results, making it difficult to assess the true effectiveness of the proposed method.
4. While the authors state that the key contribution of this work is abductive reasoning on TKGs, the proposed methods and experiments are not sufficient to support this novelty claim. The authors modify TKG data for abductive reasoning in the problem definition section, but none of the proposed methods in Section 4 appears to be specifically targeted to handle abductive reasoning on TKGs.

**Questions:**

1. It is unclear why the authors did not include static KG-based methods as baselines. If these methods are not directly applicable to TKGs, the authors should either adapt them for comparison or elaborate in the manuscript (e.g., in the related work or experimental setup) on why this is not feasible.
2. The citation for Smatch appears to be missing. The authors should also specify which version of the Smatch metric was used. Moreover, I recommend that the authors include a brief definition of Smatch in the paper to make the manuscript more self-contained.

---

> ### Author Response · Authors · 2025-11-13
>
> We sincerely thank the reviewer for the constructive and detailed feedback. Below we address each concern and clarify the misunderstandings regarding related work, temporal constraints, and baseline feasibility.
>
> R1. On missing discussion of abductive reasoning on static KGs (Bai et al., 2024; Gao et al., 2025)
>
> Thank you for pointing this out. We agree that these works deserve explicit discussion in the Related Work section, and we will add a dedicated paragraph accordingly. Although we cite Bai et al.\ and Gao et al., their problem settings differ fundamentally from ours. In the revised version, we will highlight these differences more clearly and include a conceptual comparison to better distinguish the temporal setting and demonstrate the novelty of our approach.
>
> R2. On unclear notations (Section 3) and missing definitions
>
> We thank the reviewer for pointing out this issue. Regarding the notation, $r_{ij,t}$ denotes a concrete literal, such as $r(u,v,t)$ or its negation, where the double subscript only marks its position within the DNF clause. The term $r(u,v,t)$ corresponds to the actual semantic predicate in the temporal knowledge graph, and formally $r_{ij,t} \in {, r(u,v,t),, \neg r(u,v,t) ,}$.
>
> We will add the formal definitions of the Dice and Overlap coefficients to make the manuscript self-contained. We will also clarify the semantics of answer sets in Eq.~(2) and provide additional notation examples. These updates will improve clarity without affecting the main results.
>
> R3. On the GPT-2 baseline and the concern about weak comparison
>
> As stated in Appendix~A, we represent entities, relations, and timestamps using ID tokens to avoid semantic leakage. Under this ID-level representation: (a) knowledge-enhanced language models (e.g., COMET, KG-BERT, KEPLER), (b) LLaMA or other instruction-tuned models, and (c) static KG abduction methods cannot be directly applied, as their advantages rely on rich textual semantics that are not available in our inputs.
>
> R4. On whether our method sufficiently targets abductive reasoning in TKGs
>
> We appreciate this important question. Our key design components are specifically tailored to temporal abductive reasoning:
>
> (a) \textit{Time-aware relation remapping:} each quadruple $(u, r, v, t)$ is transformed into a time-aware relation $r_t$, making every literal temporal.
>
> (b) \textit{Temporal sampling constraints:} observations and hypotheses are generated from time-indexed snapshots, and their validity depends on the corresponding snapshot. Static KG methods do not support this requirement.
>
> Q1. Why were static KG-based abduction methods not included as baselines?
>
> Our initial reasoning was that static abduction methods assume time-invariant facts. These approaches operate on triple-based KGs $(u, r, v)$ and generate hypotheses without temporal operators. In contrast, our setting requires every literal to be time-stamped, since the validity of a hypothesis depends on the specific snapshot $G_t$.
>
> Based on the reviewer’s suggestion, we agree that including a conceptual comparison with static abduction methods can further clarify the distinctions and highlight the benefits of temporal abductive reasoning. We will incorporate this discussion in the revised version.
>
> Q2. Smatch citation is missing. Which version was used?
>
> We appreciate this observation. We will include the proper citation for the Smatch metric and briefly describe it in the main text. Smatch computes the maximum matching between two AMR-style graphs based on node/edge alignment and returns an F1 score over aligned triples.
>
> We thank the reviewer again for the valuable comments. We will incorporate all suggested clarifications, expand the related work accordingly, and emphasize more clearly how temporal constraints are integrated throughout our method.

---

### Meta-Review · Area_Chair_SiEM · 2026-01-12

**Summary:**

Reviewer WYQJ concerns its unclear notations,  and the weak baseline comparison makes it difficult to assess the true effectiveness of the proposed method. The experiments are not sufficient to support this novelty claim. Reviewer 6vPf concerns its limited theoretical grounding, and some important baselines are also missing.  Reviewer TA8e  concerns that it  exhibits substantial similarity to the existing work, and  the Related Work section does not sufficiently review recent developments.

**Reviewer Concerns:**

After rebuttal, the unclear notation problem and temporal reasoning baseline problem will be addressed, while the novelty and originality may be still insufficient.

**Reviewer Scores:**

Reviewer WYQJ may maintain the original score since the baselines are a quite outdated, and the experiments are not sufficient to support this novelty claim. Reviewer 6vPf may maintain the original score since the paper lacks formal insight into why temporal constraints and the merged tokenization improve abductive inference, and it seems that the response does not provide sufficient explanations.  Reviewer TA8e may maintain the original score, since the method demonstrates a significant lack of originality and the core framework is nearly identical to that of current work.

---

### Decision · Program_Chairs · 2026-01-26

Reject